# Viral proteins as a potential driver of histone depletion in dinoflagellates

Nicholas A.T. Irwin [1,2], Benjamin J.E. Martin[1], Barry P. Young[3], Martin J.G. Browne[4], Andrew Flaus[4], Christopher J.R. Loewen[3], Patrick J. Keeling [2] & LeAnn J. Howe[1]

Within canonical eukaryotic nuclei, DNA is packaged with highly conserved histone proteins into nucleosomes, which facilitate DNA condensation and contribute to genomic regulation. Yet the dinoflagellates, a group of unicellular algae, are a striking exception to this otherwise universal feature as they have largely abandoned histones and acquired apparently viral-derived substitutes termed DVNPs (dinoflagellate-viral-nucleoproteins). Despite the magnitude of this transition, its evolutionary drivers remain unknown. Here, using *Saccharomyces cerevisiae* as a model, we show that DVNP impairs growth and antagonizes chromatin by localizing to histone binding sites, displacing nucleosomes, and impairing transcription. Furthermore, DVNP toxicity can be relieved through histone depletion and cells diminish their histones in response to DVNP expression suggesting that histone reduction could have been an adaptive response to these viral proteins. These findings provide insights into eukaryotic chromatin evolution and highlight the potential for horizontal gene transfer to drive the divergence of cellular systems.

---

[1] Department of Biochemistry and Molecular Biology, Life Sciences Institute, University of British Columbia, Vancouver V6T 1Z3 BC, Canada. [2] Department of Botany, University of British Columbia, Vancouver V6T 1Z4 BC, Canada. [3] Department of Cellular and Physiological Sciences, Life Sciences Institute, University of British Columbia, Vancouver V6T 1Z3 BC, Canada. [4] Centre for Chromosome Biology, School of Life Sciences, National University of Ireland Galway, Galway, Ireland. Correspondence and requests for materials should be addressed to N.A.T.I. (email: nickatirwin@gmail.com) or to L.J.H. (email: ljhowe@mail.ubc.ca)

The conserved organization of DNA in the eukaryotic nucleus is a paradigm in biology. Within the nucleus, DNA is bound to highly conserved protein octamers comprised of two copies of each of the four core histones: histone H2A, H2B, H3, and H4 (ref. [1]). These histones, in combination with approximately 146 bp of DNA, coalesce to form nucleosomes which act as the fundamental repeating units of eukaryotic chromatin and serve to facilitate DNA condensation[1,2]. Furthermore, histones are often post-translationally modified, especially on their intrinsically disordered N-terminal tails, leading to altered nucleosome dynamics and the recruitment of transcription, replication, and DNA repair factors[3–6]. As a result, nucleosomes play a fundamental role in genomic regulation and consequently, histones constitute some of the most highly conserved proteins known. For example, both histones H3 and H4 share roughly 90% amino acid sequence identity between yeast and humans despite around a billion years of divergence[7]. Therefore, histones contribute heavily to the growth and development of eukaryotic organisms and, given their conservation, are often viewed as a prerequisite for complex cellular life.

The dinoflagellates, a group of ecologically important unicellular eukaryotic algae, are a striking exception to the above paradigm as they have abandoned histones as their primary DNA packaging proteins[8,9]. Phylogenetic analyses have revealed that histone depletion coincided with dramatic changes in nuclear characteristics including massive genome enlargement, the emergence of liquid crystalline chromosomes, and the acquisition of apparently viral-derived proteins termed DVNPs (dinoflagellate-viral-nucleoproteins)[8,10–12]. In basal dinoflagellates, DVNPs represent the predominant basic nucleoproteins and localize to chromosomes, suggesting that they play a direct role in chromosome organization[8,11]. Accordingly, it has been hypothesized that these nucleoproteins could have been transferred from viruses to dinoflagellate progenitors with canonical chromatin and eventually replaced the majority of histones as chromatin packaging proteins.

Even though the bulk of their chromatin has diverged, dinoflagellates retain a full complement of histone genes[8,13–15]. The function of these remnant histones remains unclear, yet their low expression levels, relaxed conservation, and the presence of histone chaperones may indicate some degree of subfunctionalization to certain cellular processes, such as transcription[14]. Thus, not only the evolutionary mechanisms that drove dinoflagellate chromatin divergence but also the exact contributions of DVNPs and histones to dinoflagellate chromatin structure and function have yet to be resolved.

The above questions have remained unanswered in large part due to the technical challenges associated with studying dinoflagellate biology. In particular, a lack of genetic transformation methods and comprehensive genomic data, resulting from the size and complexity of dinoflagellate genomes, have created experimental restrictions. One way of avoiding these issues is to utilize model organisms. *Saccharomyces cerevisiae* represents a suitable model for investigating chromatin evolution because of its well-characterized and typical chromatin biology, its genetic malleability, and its well annotated genome. Therefore, to circumvent the limitations associated with dinoflagellates and gain insights into the initial transition between histone and DVNP-based chromatin, we employed an experimental evolutionary approach utilizing *S. cerevisiae* to assess how DVNP interacts with canonical eukaryotic chromatin. To this end, we found that DVNP antagonizes chromatin by localizing to histone binding sites, displacing nucleosomes, impairing transcription, and ultimately inhibiting growth. However, DVNP toxicity can be attenuated through histone depletion and cells reduce their histones following DVNP expression. These results reveal that histone depletion is an adaptive response to DVNP and emphasize the role that horizontal gene transfer, and possibly pathogenic stresses, can play in driving cellular evolution.

## Results

**DVNP enters the nucleus and impairs growth in *S. cerevisiae*.** In order to examine the interactions between DVNP and nucleosomal chromatin, we first codon optimized and synthesized *Hematodinium* sp. DVNP.5 and placed it under the control of the galactose-inducible and dextrose-repressible *GAL1* promoter (Supplementary Fig. 1). SV40 nuclear localization signals (NLS) and or three hemaglutinin (3HA) epitope tags were added to the N-terminus or C-terminus and protein expression was confirmed by immunoblot following galactose induction (Fig. 1a, b). Immunofluorescence revealed co-localization between DVNP and Hoescht stain with all constructs, suggesting that DVNP localized to the nucleus independent of the additional NLS (Fig. 1c).We also noted DVNP-dense regions associated with the nucleus (Fig. 1c). This may reflect partial nucleolar localization as nucleoli are depleted of DNA dyes and because the cationic N-terminus of DVNP could act as a general nucleolar targeting signal[16,17].

To investigate the phenotypic effects of DVNP in yeast, we performed growth assays and found that DVNP expression impaired growth (Fig. 1d), consistent with a previous report in *Toxoplasma gondii*, which is a closer relative of dinoflagellates than yeast[8]. Moreover, the addition of an N-terminal tag abrogated DVNP toxicity, either as a result of impaired function or diminished expression as the sequence composition, and therefore the immunogenicity, of the N-terminal and C-terminal 3HA tags differed (Fig. 1b, d, Supplementary Fig. 1f). In contrast, the addition of a C-terminal NLS accentuated the growth defect suggesting that toxicity may be dependent on nuclear localization (Fig. 1d).

**DVNP disrupts nucleosomal chromatin in *S. cerevisiae*.** Given the possible dependency of DVNP toxicity on nuclear localization and the capacity of DVNP to strongly and non-specifically associate with DNA in vitro[8], we hypothesized that DVNP was associated with the yeast genome. To assess this, we performed chromatin-immunoprecipitation (ChIP) using anti-HA antibodies and recovered a 17.6-fold increase in immunoprecipitated DNA in DVNP-3HA-NLS-expressing cells relative to the vector control (Supplementary Fig. 2a). To investigate the genomic localization of DVNP, we sequenced the immunoprecipitated DNA and inputs (ChIP-seq). In contrast to previous in vitro results[8], we found that rather than associating non-specifically to areas of free DNA, such as the nucleosome-depleted regions (NDRs) over promoters, DVNP was depleted at NDRs and enriched upstream and downstream of transcription start sites (TSS) (Fig. 2a). This binding profile is reminiscent of nucleosome binding, characterized by prominent −1 and +1 nucleosome peaks upstream and downstream of the TSS (Fig. 2a)[18]. These data therefore indicate that DVNP localizes preferentially to nucleosome bound regions of the genome.

The similarities in the binding profiles between DVNP and nucleosomes suggested that DVNP interacts with chromatinized DNA. To investigate whether this alters chromatin structure, we compared nucleosome profiles between DVNP-3HA-NLS-expressing cells and a vector control using micrococcal nuclease (MNase) sequencing (MNase-seq). Nucleosomal peak height and trough depths decreased in the presence of DVNP, which is indicative of nucleosome disruption (Fig. 2b)[19,20]. To assess whether nucleosome loss was DVNP-dependent, genomic windows were binned by DVNP enrichment and changes in

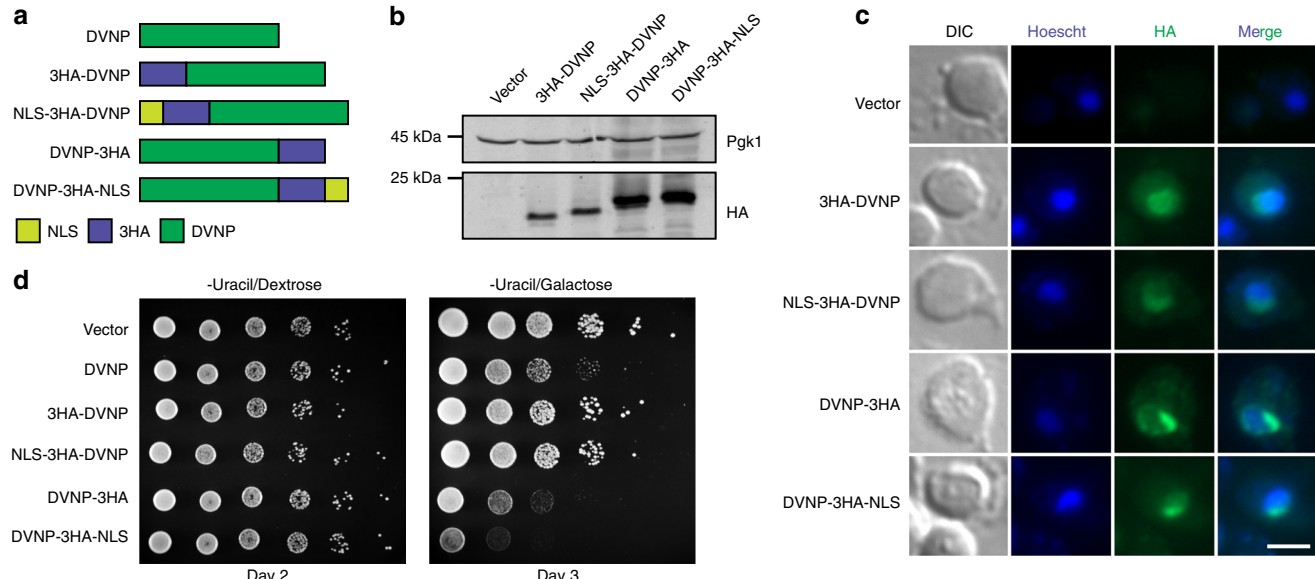

**Fig. 1** DVNP is localized to the nucleus and impairs growth in *S. cerevisiae*. **a** Schematics of DVNP constructs. **b** Immunoblot of total protein extracts from galactose-induced cells. **c** Immunofluorescence micrographs showing colocalization between the HA epitopes on DVNP (green) and Hoescht stain (blue). Scale bar, 3 μm. This experiment was repeated twice with the same results. **d** Serial dilution growth assays for cells containing indicated expression constructs. Cells were grown on selective media lacking uracil in the presence of either dextrose or galactose

nucleosome occupancy were inspected in these bins (Fig. 2c, Supplementary Fig. 2b). This revealed a negative association between DVNP enrichment and nucleosome loss that was independent of ChIP-seq inputs (Fig. 2c, Supplementary Fig. 2c). We also found that displacement predominantly occurred at the −1 and +1 nucleosomes, mirroring the localization of DVNP (Fig. 2d, Supplementary Fig. 2d). DANPOS (Dynamic Analysis of Nucleosome Position and Occupancy by Sequencing) reaffirmed that DVNP and reduced nucleosome occupancy were associated, but connections between changes in fuzziness or position and DVNP were less apparent (Supplementary Fig. 2e)[21]. We also investigated the association between nucleosome loss and stability by binning nucleosomes by their predicted occupancy, inferred from nucleosomal sequence preference (Fig. 2e, Supplementary Fig. 2f)[22]. We found that weaker nucleosomes experienced significantly greater loss than more stable nucleosomes, suggesting that nucleosomal stability prevents DVNP disruption. Overall, these data suggest that DVNP preferentially associates with nucleosomal regions of the genome and induces histone displacement.

**DVNP impairs transcription in *S. cerevisiae*.** Previous work has emphasized the importance of nucleosomes in regulating the recruitment and processivity of RNA polymerase II[20,23–26]. This led us to investigate whether DVNP adversely affects transcription by performing ChIP-seq for Rpb3, the third largest subunit of RNAP II, in DVNP-3HA-NLS-expressing and control cells. Using spike-in controls for normalization, we identified a ~35% global reduction in RNAP II occupancy that was corroborated by quantitative PCR (Fig. 3a, b, Supplementary Fig. 3a, b). We also observed a reduction of the Rpb3 peak over the TSS, consistent with the localization of DVNP (Fig. 3a, Supplementary Fig. 3a). However, the loss of Rpb3 was not dependent on transcriptional rate or DVNP abundance and was only weakly positively associated with nucleosome loss (Supplementary Fig. 3c–e).

**Histone reduction relieves DVNP toxicity in *S. cerevisiae*.** Given the deleterious effects of DVNP on cell growth, we next

wondered how an ancestral organism with canonical chromatin could have come to tolerate this protein. We therefore tested whether genetic changes could facilitate resistance to DVNP toxicity using a synthetic genetic array (SGA) analysis, whereby the relative growth of 5426 non-essential yeast deletion mutants expressing DVNP-3HA-NLS was assessed (Supplementary Data 1). Functional classification of gene deletions causing improved growth revealed chromatin and transcription associated categories as the most significant functional hits (Fig. 4a)[27]. This chromatin connection and the loss of histones in dinoflagellates led us to investigate whether histone expression altered DVNP toxicity. By analyzing 42 gene deletions previously shown to affect histone gene expression[28,29], we found that reducing and increasing histone production relieved and exacerbated DVNP toxicity, respectively (Fig. 4b). Moreover, we identified a loss of total histones H3, H4, and H2B following DVNP induction in wild type cells (Fig. 4c–f). These data suggest that histone reduction is an adaptive response to DVNP toxicity and that cells cope with DVNP by maintaining a lower abundance of histones.

Of all the genes inspected, only four exceptions were noted. In particular, deletions of *SPT21*, a sequence specific histone gene activator, and *HHF2*, one of the two genes encoding histone H4, increased toxicity in the SGA screen (Fig. 4b)[29,30]. However, neither of these deletions were detrimental when manually assessed and a newly generated *spt21Δ* mutant in a different strain background relieved toxicity despite DVNP levels being unchanged (Supplementary Fig. 4a–e). Furthermore, deletion of components of the TRAMP (Trf4/Air2/Mtr4p polyadenylation) complex and Xrn1, which negatively regulate histone levels[31,32], improved growth (Fig. 4b). However, these proteins are involved in general RNA degradation[33,34], so their removal could promote RNA stability and relieve problems associated with transcriptional defects.

To reaffirm our results, we also assessed whether histone depletion could be a non-specific adaptive response to the over-expression of any toxic exogenous or endogenous nuclear protein. To examine this, we performed the same analyses with published SGA analysis data for two proteins expressed from the same *GAL1* promoter, TDP-43 and Hho1. TDP-43 is a toxic

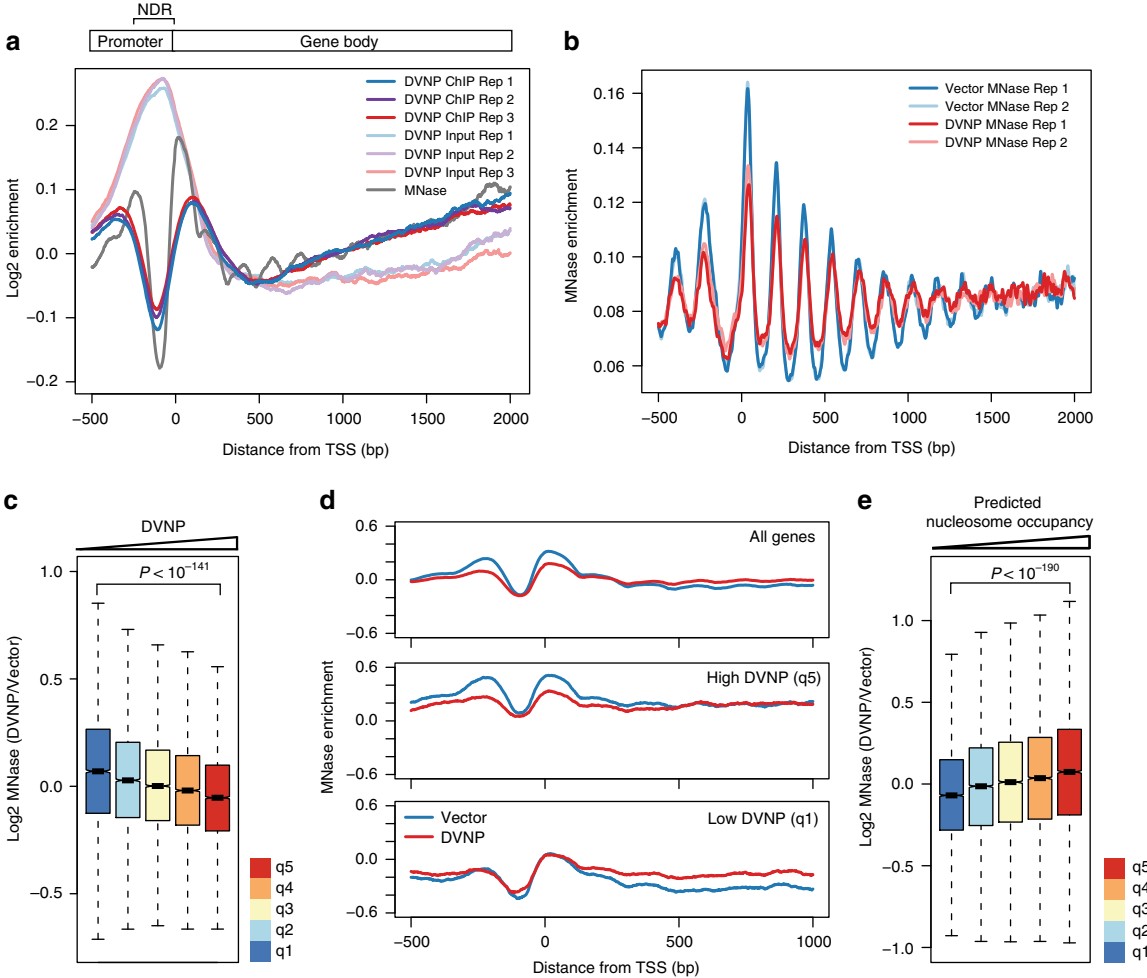

**Fig. 2** DVNP binds nucleosomal regions and induces histone loss in *S. cerevisiae*. **a** Average gene plot showing the relative enrichment of DVNP chromatin immunoprecipitates (ChIPs), and ChIP inputs from DVNP-3HA-NLS-expressing cells. Three biological replicates are shown for the ChIP and input. Also shown is the enrichment profile for MNase-digested control cells. A schematic of an average gene is shown above, with the nucleosome-depleted region (NDR) labeled. DVNP ChIP and input replicates were combined in all subsequent analyses. **b** Average gene plot showing MNase digested DNA from two biological replicates obtained from DVNP-3HA-NLS-expressing and control cells. Enrichment was calculated from read midpoints and Gaussian smoothed with a standard deviation of 4. MNase-seq replicates were combined in all further analyses. **c** Genome-wide relationship between DVNP and nucleosome loss. Five hundred base pair genomic windows were binned into 20% DVNP quintiles (q1-5, see Supplementary Fig. 2b) and nucleosome loss is shown in these bins (*n* = 9659). Outliers are shown in Supplementary Fig. 2g. **d** Average gene plots showing MNase profiles for DVNP-3HA-NLS-expressing and control cells. Averages were calculated from all genes (top panel), or genes with the top (q5, middle panel) or bottom (q1, bottom panel) 20% DVNP enrichment over the gene body (see Supplementary Fig. 2d). **e** Genome-wide association between predicted nucleosome occupancy and nucleosome loss due to DVNP expression. Nucleosome bound sites were binned into 20% predicted occupancy quintiles (q1-5, see Supplementary Fig. 2f) and nucleosome loss is shown in these bins (*n* = 13,509). Outliers are shown in Supplementary Fig. 2h. All *P*-values were calculated using two sided Welch's *t*-tests. Box plot notches represent an estimate of the 95% confidence interval of the median

mammalian DNA-binding protein whereas Hho1 is the yeast homolog of histone H1, which resembles the size and basicity of DVNP (Supplementary Fig. 4f, g)[35,36]. Although both of these proteins significantly impair growth when over-expressed[35,36], no similar effect of histone levels on the TDP-43 or *HHO1* over-expression phenotype was observed, revealing that DVNP's genetic interactions are not ubiquitous.

## Discussion

Here we sought to investigate how the dinoflagellate DNA-binding protein, DVNP, interacts with the canonical chromatin of yeast to gain insights into dinoflagellate chromatin divergence. To this end, we showed that DVNP interacts antagonistically with nucleosomal chromatin, causing histone displacement, transcriptional impairment, and growth inhibition, but that histone

reduction partially mitigates this toxicity. It is possible that histone depletion relieves toxicity through transcriptional up-regulation, as is seen in ageing yeast with reduced histones[37], or by reducing excess displaced histones which are cytotoxic[38]. In either case, it leads to a model for the origin of dinoflagellate nuclear organization based on a stepwise increase in DVNP and corresponding depletion of histones. On one hand, forced exposure to DVNP, such as during viral infection, may have prompted histone depletion as a mechanism for limiting DVNP toxicity. Alternatively, DVNP may have been introduced during a transiently histone-depleted stage, for example following histone dilution in the wake of genome expansion. In the first instance, histone depletion would be a direct response to the most deleterious effects of DVNP, which in turn would open the door to more DVNP binding ultimately resulting in a large-scale displacement of histones by DVNP. In the second instance, it is

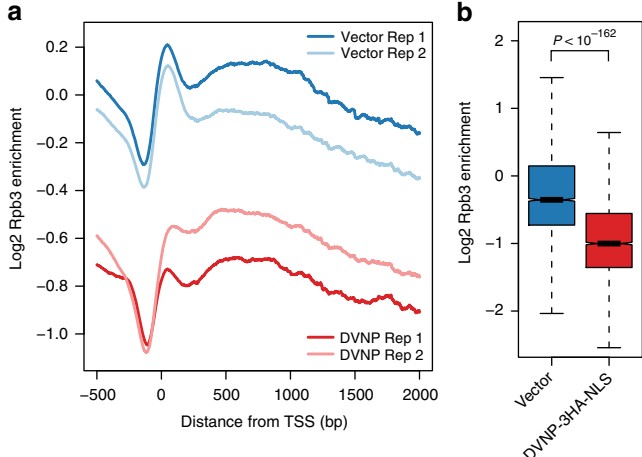

**Fig. 3** DVNP expression results in transcriptional impairment in *S. cerevisiae*. **a** Spike-in normalized average gene plot showing the enrichment of two biological replicates of Rpb3 ChIPs from DVNP-3HA-NLS-expressing and control cells. **b** Differences in Rpb3 following DVNP expression. Values represent the mean Rpb3 occupancy calculated over gene bodies after averaging biological replicates ($n = 4793$). Outliers are shown in Supplementary Fig. 3f. The *P*-value was calculated using a two sided Welch's *t*-test. Box plot notches represent an estimate of the 95% confidence interval of the median

possible that DVNP had a mild short-term benefit in an already histone depleted system, and its presence may have prevented the re-colonization of chromatin by histones over time. In either case, it would appear that something about the underlying biology of the ancestral dinoflagellate made it possible for the invasion of DVNP to lead to a progressively shifting balance between histones and DVNP, over time resulting in a functional replacement by DVNP as the major genome packaging protein.

Although DVNP is likely of viral origin given the homology it shares with proteins in algae-infecting viruses[8,39], the actual source of DVNP remains to be clarified. No virus of this kind has been found in dinoflagellates, although the diversity of phycodnaviruses is not well sampled, and our model infers such an infection in the distant evolutionary past. But given the deleterious effects of DVNP expression, the most likely context for DVNP to be acquired would be pathogenesis, since this gives a powerful selective force for the depletion of histones, which in normal contexts would itself be deleterious. Moreover, viruses frequently utilize chromatin effectors during pathogenesis to manipulate host processes and defenses. For example, foot-and-mouth virus protease 3C and adenovirus protein VII disrupt cellular expression and signaling by interacting with host nucleosomes[40,41]. Despite this, other sources of DVNP are also possible. Ancestral dinoflagellates could have been less susceptible to DVNP, facilitating passive acquisition from a virus, food, or commensal symbiont. However, DVNP is unknown in cellular genomes outside dinoflagellates, making this less likely. The

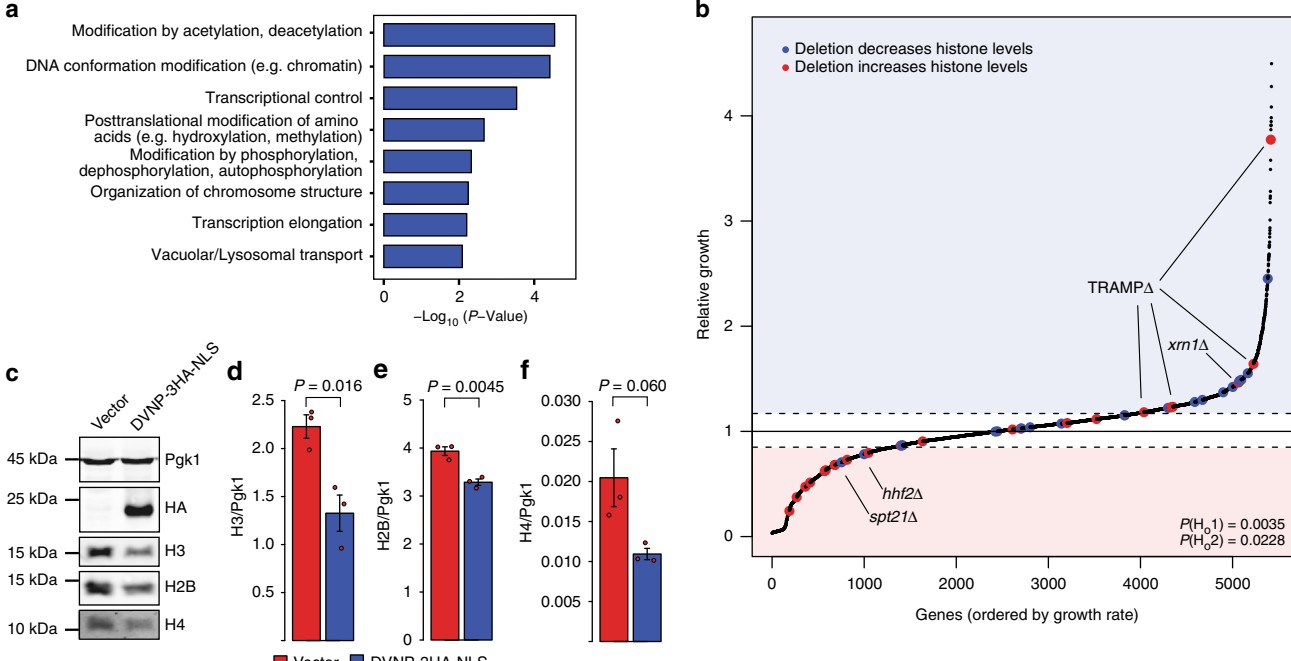

**Fig. 4** Histone loss is an adaptive response to DVNP toxicity in *S. cerevisiae*. **a** Significant MIPS functional classifications of gene deletions that relieved DVNP toxicity in the SGA screen. **b** Relative growth of ~5500 non-essential deletion strains expressing DVNP-3HA-NLS from a *GAL1* promoter. Gene deletions known to increase and decrease histone expression are shown in red and blue, respectively. Two null hypotheses were rejected by $\chi^2$ test: H$_0$1: gene deletions that affect histone levels are randomly distributed and, H$_0$2: gene deletions that increase and decrease histones are enriched below and above the growth thresholds by chance. Dashed lines denote positive and negative growth thresholds. Exceptions to the trend have been labeled. See Supplementary Data 1 for a full list of genes. **c** Immunoblot of total protein extracts from DVNP-3HA-NLS-expressing and control cells. **d-f** Quantified immunoblots showing histone levels in DVNP-3HA-NLS-expressing and control cells ($n = 3$ biological replicates). Histone signal was normalized to the loading control, Pgk1, which remained constant between conditions. These experiments were repeated three times with similar results. *P*-values were obtained by two sided Welch's *t*-test. Error bars represent the standard error of the mean (SEM)

activity of DVNP in modern viruses infecting their current hosts would presumably shed some light on these possibilities.

Why DVNP is not observed in other organisms is an interesting question. DVNP-related proteins have been identified in viruses infecting diverse algae such as the stramenopile *Ectocarpus siliculosus* and green alga *Micromonas*, yet these organisms have not acquired DVNP and their histone-based chromatin is unaffected[8,39]. This may simply be due to the low probability of initiating such drastic change to chromatin structure. However, it may also be that other unique aspects of dinoflagellate biology "preconditioned" the system such that this radical reaction to the introduction of DVNP was more likely. For example, the tree of dinoflagellates shows that gene expression using specialized transsplicing mRNAs predates the rise of DVNP[42–44]. This suggests that gene expression in the dinoflagellate ancestor was already very unusual and that control of expression had shifted from transcriptional to post-transcriptional mechanisms. Genetic systems with such characteristics could react very differently to a perturbation such as the introduction of DVNP and histone displacement.

Despite the antagonism we observe between DVNP and histones, the retention of histone genes in dinoflagellates suggests that these proteins still have some role in dinoflagellate chromatin regulation. Here we find that the replacement of histones with DVNP results in a net loss of RNAP II occupancy, indicating that the yeast transcriptional machinery is less equipped to deal with DVNP than with nucleosomes. Recent bioinformatic analyses have revealed that dinoflagellate histones have relaxed selection over heterochromatin-associated modification sites, whereas activating sites have been conserved, suggesting that dinoflagellates may lack nucleosomal heterochromatin[14]. This role is seemingly filled by DVNP and may be accomplished through its enhanced ability to repress transcription relative to histones. We also showed that DNA with a high GC content, which is predicted to form more stable nucleosomes, was more resistant to nucleosome displacement by DVNP[22,45]. Interestingly, dinoflagellates have a strong GC codon bias, which may promote nucleosome stability in open reading frames[46]. This, together with the known conservation of activating histone modifications and histone chaperones in dinoflagellates[14], suggest that nucleosomes and DVNP may function in euchromatic and heterochromatic environments, respectively. If nucleosomes have subfunctionalized in dinoflagellates, then biochemical investigations into the activities of dinoflagellate histones may provide unique insights into the roles histones play in diverse eukaryotes by highlighting some of their specific functions, beyond bulk chromatin condensation.

## Methods

**Plasmids and yeast strains**. DVNP.5 from the dinoflagellate *Hematodinium* sp. (accession number: AFY23231.1) was codon optimized for expression in *S. cerevisiae* and synthesized by GenScript into a pUC57 vector[8]. DVNP was then amplified with the addition of 5′ SpeI and NdeI restriction sites and a 3′ XmaI site using high fidelity Kapa TaqReadyMix (Kapa Biosystems) polymerase chain reaction (PCR). The DVNP amplicon was then cloned into a pRS416 expression vector containing a *GAL1* promoter (GAL1pr) using SpeI and XmaI restriction enzymes (New England BioLabs (NEB))[47,48] (Supplementary Fig.1a–e). To add a 3HA epitope tag to the C-terminus of the protein, DVNP was cloned into a pRS416 vector containing a C-terminal 3HA tag[36]. In contrast, a 3HA tag was added to the N-terminus of DVNP by performing gap repair on an NdeI (NEB)-digested DVNP plasmid using a 3HA gene block synthesized by Integrated DNA Technologies[49]. The discrepancies between these two methods led to sequence variation between the N-terminal and C-terminal 3HA tags (see Supplementary Fig. 1f). Lastly, SV40 NLS were added to the N and C termini of DVNP by PCR[50]. All plasmids were confirmed by sequencing which was conducted by the Nucleic Acid/Protein Service Unit at the University of British Columbia.

Plasmids were transformed into yeast using a lithium acetate-based protocol[51]. All yeast strains used in this study are listed in Supplementary Table 1. *SPT21* was deleted from the wild type strain (FY602 (ref.[52])) by targeted homologous

integration using an amplicon containing the *HIS3* marker gene flanked by the ends of the *SPT21* gene[53]. Deletion was confirmed by PCR using primers directed within the marker gene and upstream of *SPT21*.

**Total protein extraction and immunoblotting**. Yeast strains containing expression vectors were grown to saturation at 30 °C in synthetic dropout media lacking uracil and supplemented with 2% dextrose. To induce DVNP expression, cells were collected by centrifugation at 3000×g for 3 min and then washed twice and ultimately resuspended in the same dropout media containing 2% galactose. Cultures were grown in galactose for 16 h at 30 °C to an optical density ($OD_{600}$) of $0.8 ± 0.1$ prior to collection. Cell numbers were normalized by $OD_{600}$ and total protein was isolated using a mild 0.2 M NaOH alkali treatment[54].

Protein samples were heated for 5 min at 95 °C and separated using 15% SDS-PAGE (sodium dodecyl sulfate–polyacrylamide gel electrophoresis). Following electrophoresis, gels were equilibrated in an SDS buffer (62.5 mM Tris pH 6.8, 2.3% SDS) for 30 min prior to being transferred to a nitrocellulose membrane in an ethanolamine transfer solution (0.15% ethanolamine, 0.017 mM glycine, 20% methanol). Transfer efficiency and equal protein loading was confirmed by ponceau staining prior to blocking in 2% powdered milk in PBS-T (0.68 M NaCl, 13.4 mM KCl, 50 mM $Na_2HPO_4$, 8.8 mM $KH_2PO_4$ pH 7.4, 1% Tween-20) for 2 h at room temperature. Membranes were incubated with the following primary antibodies: HA (Roche, High affinity 3F10 clone, 1:2500, 16 h, 4 °C), Pgk1 (Novex, 459250, 1:10,000, 1 h, 20 °C), H3 (Genscript, rabbit polyclonal raised to antigen CKDIKLARRLRGERS, 1:5000, 16 h, 4 °C), H4 (Abcam, ab31830, 1:2000, 16 h, 4 °C), or H2B (Active Motif, 39237, 1:2000, 16 h, 4 °C). Following primary antibody incubation, membranes were washed three times in PBS-T and incubated with anti-rat (LiCOR, 926–32219), anti-mouse (LiCOR, 926–32221), and or anti-rabbit (LiCOR, 926–32210) secondary antibodies at 1:15,000 dilutions for one hour at room temperature. Membranes were washed in PBS-T for 25 min and imaged using a LiCOR Odyssey imaging system. Protein quantification was performed using LiCOR Odyssey Infrared Imaging software v3.0. Full gel images are shown in Supplementary Fig. 5.

**Immunofluorescence microscopy**. Immunofluorescence was conducted using a previously developed protocol[55], with some modifications. Cells constitutively expressing DVNP from pRS416 vectors containing *HHT2* promoters[36] were grown in synthetic dropout media lacking uracil to an $OD_{600}$ of 0.4 before being harvested and fixed in 3.7% formaldehyde for 1 h at 25 °C. Fixed cells were pelleted at 9000×g for 30 s before being washed twice in SK buffer (1 M sorbitol, 50 mM KPO₄, pH 7.5) and stored at 4 °C for 48 h. Fixed cells were then applied to poly-L lysine (Sigma) coated slides and allowed to settle for 5 min. The cell solution was then aspirated and the slide was washed twice with SK buffer prior to being submerged in a −20 °C methanol bath and −20 °C acetone bath for 6 and 3 min, respectively. Non-specific sites were blocked with 3% bovine serum albumin (BSA) in PBS for 20 min. The slides were then incubated with 1:100 HA antibody (Roche, High affinity 3F10 clone) diluted in 3% BSA PBS for one hour at 37 °C in a humidified chamber. Following this, the slides were washed with PBS and incubated as above with 1:2000 fluorescein conjugated anti-rat antibody for 45 min. Slides were finally washed again and mounted with fluoromount aqueous mounting media (Sigma) containing 2.5 μg/mL Hoechst stain. Micrographs were acquired on a Zeiss Axio Observer inverted microscope equipped with a Zeiss Colibri LED illuminator and a ZeissAxiocam ultrahigh-resolution monochrome digital camera Rev 3.0. Immunofluorescent images were analyzed using Zeiss ZEN software v2.1 and ImageJ.

**Chromatin immunoprecipitation**. ChIP experiments were performed based on previously outlined protocols[56]. Cells were grown as described above (see Total protein extraction and immunoblotting) before being fixed in 1% formaldehyde for 30 min at room temperature. Excess formaldehyde was quenched with 125 mM glycine for 15 min and then cells were pelleted at 3000×g for 3 min at 4 °C and washed with cold PBS. After three washes, cells were normalized to 40 OD units (ODU) before being frozen at −80 °C.

Following thawing, cells were resuspended in lysis buffer (50 mM HEPES pH 7.5, 140 mM NaCl, 0.5 mM EDTA, 1% Triton X-100, 0.1% Na-deoxycholate, 1 mM phenylmethanesulfonyl fluoride, and 1X Protease inhibitor cocktail (Roche)) and lysed by bead beating. Cell lysates were pelleted at 15,000×g for 30 min at 4 °C, washed and resuspended in lysis buffer, and sonicated for 30 cycles of 30 s on/30 s off at high power at 4 °C using a Biorupter sonicator (Diagenode). Sonicated lysates were then pre-cleared with protein G conjugated magnetic beads (Dynabeads, Thermo Fisher) for 1 h at 4 °C. After clearing, 6% of the lysate was collected as "input" and 1:400 anti-HA (Roche, High affinity 3F10 clone) antibody or 1:1250 anti-Rpb3 (Abcam, ab81859, monoclonal clone 1y26[1y27]) antibody was added prior to 16 h of rotation at 4 °C. Antibodies were extracted using protein G conjugated magnetic beads (Dynabeads, Thermo Fisher) for 4 h at 4 °C and the beads were subsequently washed twice with lysis buffer, twice with lysis buffer supplemented with 500 mM NaCl, twice with lithium buffer (10 mM Tris-HCl pH 8.0, 250 mM LiCl, 0.6% NP-40, 0.5% Na-deoxycholate, 1 mM EDTA pH 8), and once with TE buffer (10 mM Tris-HCl pH 8.0, 1 mM EDTA). Immunoprecipitates were eluted with elution buffer (10 mM Tris-HCl pH 8.0, 1 mM EDTA pH 8, 1% SDS, 150 mM NaCl, 5 mM DTT) at 65 °C and then treated with 80 μg/mL

proteinase K at 65 °C for 16 h and 300 µg/mL RNase A at 37 °C for 2 h. For the Rpb3 ChIP, prior to DNA purification, spike-in DNA (10:1 ($2 \times 10^{-4}$: $2 \times 10^{-5}$ ng/µL) spike-in 1:2) was added to a 1:400 and 1:3.33 dilution in the ChIPs and inputs, respectively (Supplementary Table 2). DNA purification was performed using a Qiagen Minelute PCR purification kit or by phenol:chloroform:isoamyl extraction. DNA fragmentation and concentration were assessed using a 1% agarose gel containing syto60 dye (Invitrogen) and a high sensitivity Qubit fluorometer (Thermo Fisher), respectively.

**MNase digestion**. MNase digestions were performed as described previously[36]. Cells were grown as for ChIP (see Chromatin immunoprecipitation) before being normalized to 25 ODUs. Cells were resuspended in 1 M sorbitol, 5 mM β-mercaptoethanol and 10 mg/mL zymolyase prior to being incubated at 37 °C for 10 min. Spheroplasts were washed in 1 M sorbitol, twice in spheroplast digestion buffer (SDB: 1 M sorbitol, 50 mM NaCl, 10 mM Tris pH 8, 5 mM MgCl₂, 1 mM CaCl₂, 1 mM β-mercaptoethanol, 0.5 mM spermidine, 0.075% NP40) and resuspended in SDB before being digested with MNase for 2 min. Digestions were stopped with 5 mM EDTA and 1% SDS and crosslinks were reversed by overnight incubation at 65 °C. Proteinase and RNase treatment as well as DNA fragmentation assessments and concentration were done as above (see Chromatin immunoprecipitation).

**ChIP-quantitative PCR**. ChIP-quantitative PCR was performed using previously developed protocols[57]. In particular, Rpb3 ChIP eluates (see Chromatin immunoprecipitation) were diluted by a factor of 20 and quantified by quantitative PCR (qPCR). qPCR reactions were performed in technical triplicate, using SYBR green for detection in an Applied Biosystems StepOnePlus Real-Time PCR System, and quantified against a standard curve of genomic DNA. Primers used for qPCR are listed in Supplementary Table 3.

**Sequencing and bioinformatic analysis**. Sequencing libraries were constructed using 2 ng of DNA using a low-input protocol[58]. Briefly, samples were end repaired (1X T4 DNA ligase buffer (NEB), 0.4 mM dNTP mix, 2.25 U T4 DNA polymerase (NEB), 0.75 U Klenow DNA polymerase (NEB), and 7.5 U of T4 polynucleotide kinase (NEB), incubated at room temperature for 30 min), A-tailed (1X NEB buffer 2, 0.4 mM dATP, and 3.75 U of Klenow (exo-) (NEB) incubated at 37 °C for 30 min), ligated to adapters (1X Quick DNA ligase buffer (NEB), 1 mM Illumina PE adapters, and 1600 U Quick DNA-ligase (NEB), incubated at room temperature for 1 h) and PCR amplified (1X NEBNext master mix (NEB) and 0.4 µM indexed primers (Illumina)) using 12 PCR cycles with a 65 °C annealing temperature and a 30 s extension time. DNA was purified between each step using two volumes of NucleoMag NGS DNA purification beads (Macherey–Nagel) except after adapter ligation and PCR amplification where 0.8 volumes were used to facilitate size selection. Library yield and size distribution was assessed using a high sensitivity Qubit fluorometer (Thermo Fisher) and an Agilent Tape Station, respectively.

Libraries were pooled and size selected on a 2% agarose gel to between 100 and 1000 bp. Pooled libraries were then sequenced using either 80-bp paired end reads on an Illumina MiSeq using a v3 reagent kit (DVNP ChIPs, inputs, and MNase-seq) or on an Illumina HiSeq with 100 bp paired end reads using a HiSeq SBS v4 reagent kit (Rpb3 ChIPs and inputs). FASTQ files were initially assessed using FastQC v0.11.4 prior to being aligned to saccer3, the most recent build of the yeast genome (released February 3, 2011; downloaded from http://www.yeastgenome.org), using the Burrows Wheeler aligner algorithm v0.7.13 ((refs.[59,60]). Samtools v0.1.19 was then used to filter out mapped reads with mapping quality scores less than 10 ((ref.[61]). Sequence fragment sizes were filtered to exclude excessively large and small fragments as inferred from fragment size distributions. Subsequent analyses and statistics were performed using the Java Genomics toolkit (downloaded from http://palpant.us/javagenomics-toolkit/), DANPOS v2 (ref.[21]), and R v3.4.0.

Average gene profiles were obtained by averaging the sequencing coverage, which was normalized to the average genomic coverage, at each base, 500 bp upstream and 2500 bp downstream of the transcription start site (as defined by simultaneous mapping of RNA ends by sequencing (SMORE-Seq)[62]) of 4793 genes. Genes were included in these calculations until their polyadenylation sites were reached (as defined by SMORE-Seq[62]). For the Rpb3 ChIP calculations, genes were included until 300 bp from their polyadenylation sites due to large peaks at the 3′ ends of many genes that skewed quantification. Genome-wide analyses were performed by either dividing the genome into 500 bp windows with 250 bp steps or by calculating occupancy over nucleosomal sites[63]. To avoid DNA accessibility bias, all correlations were observed in and out of input-controlled bins. With regards to box plots, boxes span from the first to third quartile with whiskers extending 1.5 times the interquartile range (IQR). Black bars represent the median and notches represent an approximation of the 95% confidence interval and extend ±1/58 IQR/sqrt($n$).

**Synthetic genetic array analysis**. SGA analysis was carried out using a ROTOR colony manipulation robot (Singer Instruments) in combination with the non-essential yeast deletion array as previously outlined[64–66]. The SGA starting strain, Y7093, was transformed with the pRS416-GAL1pr-DVNP-3HA-NLS plasmid and mated with the deletion array. Diploids were selected using YPD (1% yeast extract,

2% peptone, 2% dextrose) supplemented with 0.25 mg/mL G418 and 0.1 mg/mL nourseothricin and sporulated on depleted media (1% KOAc, 0.5% yeast extract, 0.5% dextrose, 0.001% sporulation amino acid mix, 2% agar, 0.25 mg/mL G418) for 11 days at 30 °C. Double mutant haploids were then selected and cultured on germination media (0.7% yeast nitrogenous base without ammonium sulfate, 0.2% complete supplement mixture lacking arginine, lysine, histidine, and uracil, 2% dextrose, 2% agar, 0.05 mg/mL thialysine, 0.05 mg/mL canavanine, 0.25 mg/mL G418 and 0.1 mg/mL nourseothricin). The resulting strains were then plated onto either germination media containing 2% galactose and 2% raffinose (experimental plates) or germination media containing 2% galactose, 2% raffinose, and 2 mg/mL 5-fluoroorotic acid, a drug which selects for loss of *URA3*-based plasmids (control plates). Plates were imaged on a flat-bed scanner and colony size and relative growth were quantified and analyzed using Balony v1.2.1 using the default settings[67]. Default thresholds for growth and lethality were set and resulting mutants above the rescue threshold were inspected using FunSpec[68]. Functional categories were assigned using Munich Information Centre for Protein Sequences (MIPS) functional classifications[27].

**Data availability**. The ChIP-seq and MNase-seq data sets have been deposited in the Gene Expression Omnibus under accession number GSE102280. The SGA analysis data is available in Supplementary Data 1.

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

## Acknowledgements

We acknowledge Savrina Manhas, Dr. Vivien Measday, Michelle Moksa, Dr. Martin Hirst, James Kwan, Dr. Corey Nislow, and Dr. Dolph Schluter for providing assistance. This work was supported by a Natural Sciences and Engineering Research Council (NSERC) Discovery Grant awarded to L.J.H. N.A.T.I and B.J.E.M. were supported by graduate student fellowships from NSERC.

## Author contributions

N.A.T.I. and L.J.H. conceived the study. N.A.T.I., B.J.E.M., and L.J.H. performed the experiments. A.F., C.J.R.L., P.J.K., and L.J.H. provided supervision and financial support. B.P.Y and M.J.G.B. provided materials and technical assistance. N.A.T.I., P.J.K., and L.J. H. wrote the manuscript with input from all authors.

## Additional information

**Competing interests:** The authors declare no competing interests.

