## [Peer Review File · Nature Communications]

Reviewers' comments:

Reviewer #1 (Remarks to the Author):

The study used yeast as a model to express dinoflagellate/viral nucleoprotein (DVNP) and found its toxic effect to the host. It also confirms the previous (initial) result conducted in a dinoflagellate's close relative, apicomplexan, that this protein localizes to the nucleus and binds to DNA. Using CHIP-seq and enzyme digestion, the authors further demonstrated that the binding hotspots are histone-binding sites. Genetic manipulation also revealed that depletion of native histones lessened the toxicity of expressed DVNP. The mapping of DVNP binding site and the finding of functional antagonism between DVNP and histones constitute the most interesting new results. The work was nicely done and I have no technical concerns.

My only major concern regards the interpretation of the results as evidence of viral challenge being the driver of histone displacement by DVNP. Several important points cast questions on that conclusion. Firstly, while it is probable the toxic effect seen in the host yeast would occur also in dinoflagellates, the result shows what may happen after DVNP genes have been acquired and become functional, but not about how the gene may have been acquired. Because this gene is highly similar to a phycodnavirus protein gene, horizontal gene transfer from a viral origin is indeed the best explanation. Similar horizontal gene transfer has been seen in dinoflagellates: bacterial histone like proteins (I and II), which have also been thought to replace histones in dinoflagellates; form II Rubisco apparently originated from anaerobic bacterial origin. Therefore, dinoflagellates probably have received genes from viruses and bacteria. However, it is unclear whether it was from ingested food or infection, or through a third-party mediator. Secondly, the acquisition of DVNP gene did not lead to loss of histone genes in dinoflagellates, but only represses the expression and displaces (apparently) the DNA-binding function of histones. The inability to account for why and how this happens places the evolutionary process leading to DVNP acquisition on the discussion table despite the elegant experimental results reported in the manuscript. Thirdly, infection of dinoflagellates by the phycodnavirus has never been reported, yet other algal hosts are known, such as *Ectocarpus siliculosus*, and *Feldmannia* sp. Yet no acquisition of this viral gene and subsequent histone displacement have seemed to occur in those hosts. This also raises a question if viral infection can be a sufficient driving force.

Reviewer #2 (Remarks to the Author):

This is a very clever manuscript by Irwin and colleagues who take the original observation of the mutually independent phylogenetic patterns of DVNPs and histones in dinoflagellates (Current Biology 2012) and ascribe a causality to it using yeast experiments. This is a very key finding.

The authors conclude that DVNP expression in yeast is toxic, that it displaces nucleosomes

and affects RNA Polymerase recruitment, and most interestingly, that loss of histone genes or histone gene expression can exacerbate these toxic effects.

In many aspects this is the best set of experiments one can perform to describe the mysterious loss of histones and canonical chromatin in dinoflagellates. I find the arguments compelling except for one point that is much more of a discussion point than anything else.

Character displacement (DVNPs for histones) must have proceeded through a deleterious phase if the authors are correct. The more deleterious, the greater the intensity of selection for loss of chromatin genes. But if this were the case, it also holds that the more deleterious, the less likely the event would have occurred at all, or gone to fixation in a species.

To complete the argument, it might be worth speculating why the DVNP invasion was permitted in spite of its instantly deleterious consequence. After all it is unlikely that the viral invasion would have been complete in the species. As a thought or actual experiment, is it possible for the authors to assess the opposite i.e. is loss of histone gene expression (which is also deleterious in yeast) exacerbated by DVP expression. Could such an event be possibly related to the locking down of the transiently histone-less state in dinoflagellates?

---Harmit S. Malik

Reviewer #3 (Remarks to the Author):

In this paper, the authors study dinoflagellate-viral-nucleoproteins (DVNPs) interactions in chromatin. The dinoflagellates are unicellular algae and have nuclear chromatin devoid of histone proteins. Histones are replaced by these DVNP proteins which are also relatively small basic protein with >25% lysine residues and about 10% arginine residues, very similar to the basic amino acid content of histones. To study the DVNP protein interactions in chromatin, the authors insert the gene for DVNP into a strain of *Saccharomyces cerevisiae*, express it there, and then assess whether these proteins become chromatin bound and whether they have the properties of chromatin with regular histones. Figure 1 shows that the DVNP proteins localize to nuclei in yeast cells, but impairs growth. In figure 2, the authors show in a genome wide fashion that DVNPs are associated with the yeast chromatin as complexes similar to nucleosomes by histone displacement. And Figure 3 shows the effect on gene expression of the expression of the DVNPs in yeast cells. Figure 4 shows that there is a histone loss in an adaptive response to DVNP. To interpret this directly, more DVNP, less histone. The big question remains, why? Can the effect be titrated with different proteins?

Since DVNP is a basic protein comparable to histones, it is not surprising that these effects are registered in yeast. Maybe an additional control experiment to do, besides TDP-43 which is only 10% basic amino acids, would be to also add in a different equally basic protein such as a protamine, or some other proteins of similar length which are lysine and/or arginine

rich.

Minor point:

Page 6: Line 116: Fig 3f should be Fig 4f.

Reviewer #4 (Remarks to the Author):

The manuscript aims to address the mechanism with which the nucleosomeless status of the dinoflagellates evolved.

1. The description seems to be confused as to whether the data wish to address nucleosomeless or histoneless;
2. "Histone loss and histone depletion" was repeatedly mentioned;
This is incorrect as there are histone expression in dinoflagellates
[ref 28 and Roy and Morse 2012, Plos One <https://doi.org/10.1371/journal.pone.0034340>]
3. DVNP expression in yeast, and in previous experiment in *Toxoplasma*,
Apparently had no mentioned effect on the genomic copies of histones; therefore the ms did not address the question it set out to address
A better model could have been in ciliates, as the both apicomplexans and dinoflagellates were evolved from ciliates
Expression of a DNA-binding protein with higher affinity to DNA binding sites presumably would produce similar effects (e.g. protamines)

RESPONSE TO REVIEWERS

Reviewer comments are shown in black, author responses are shown in bolded blue.

Reviewer #1 (Remarks to the Author):

The study used yeast as a model to express dinoflagellate/viral nucleoprotein (DVNP) and found its toxic effect to the host. It also confirms the previous (initial) result conducted in a dinoflagellate's close relative, apicomplexan, that this protein localizes to the nucleus and binds to DNA. Using CHIP-seq and enzyme digestion, the authors further demonstrated that the binding hotspots are histone-binding sites. Genetic manipulation also revealed that depletion of native histones lessened the toxicity of expressed DVNP. The mapping of DVNP binding site and the finding of functional antagonism between DVNP and histones constitute the most interesting new results. The work was nicely done and I have no technical concerns.

My only major concern regards the interpretation of the results as evidence of viral challenge being the driver of histone displacement by DVNP. Several important points cast questions on that conclusion.

Firstly, while it is probable the toxic effect seen in the host yeast would occur also in dinoflagellates, the result shows what may happen after DVNP genes have been acquired and become functional, but not about how the gene may have been acquired. Because this gene is highly similar to a phycodnavirus protein gene, horizontal gene transfer from a viral origin is indeed the best explanation. Similar horizontal gene transfer has been seen in dinoflagellates: bacterial histone like proteins (I and II), which have also been thought to replace histones in dinoflagellates; form II Rubisco apparently originated from anaerobic bacterial origin. Therefore, dinoflagellates probably have received genes from viruses and bacteria. However, it is unclear whether it was from ingested food or infection, or through a third-party mediator.

This is an important point and we agree that the evidence for viral challenge is lacking and that it was overemphasized. Our data does not reveal the exact vector by which DVNP was obtained by dinoflagellates but rather suggests what might have happened prior to or upon its acquisition. We acknowledge that the exact order of events leading to DVNP acquisition may never be precisely known (as with many evolutionary questions) and accordingly, we have changed the title of the manuscript to make it more conservative ("Viral proteins as a potential driver of histone depletion in dinoflagellates"). We have also added a section to the Discussion regarding the origins of DVNP (see lines 186-211). There we mention that because of the deleterious effects of DVNP and the abundance of chromatin manipulating virulence factors used by viruses, viral challenge and adaptation through histone depletion may explain where DVNP came from. However, we also describe how reduced

susceptibility to DVNP could have allowed DVNP to be obtained more passively via food or a third party mediator.

Secondly, the acquisition of DVNP gene did not lead to loss of histone genes in dinoflagellates, but only represses the expression and displaces (apparently) the DNA-binding function of histones. The inability to account for why and how this happens places the evolutionary process leading to DVNP acquisition on the discussion table despite the elegant experimental results reported in the manuscript.

We agree that DVNP did not cause the loss of histone genes in dinoflagellates, because dinoflagellates actually have not lost their histone genes (see Marinov and Lynch 2015, ref. 14). They appear to have simply down regulated their histones relative to the amount of DNA, which is consistent with our results (see Fig. 4c-f). This has been clarified in the text (see lines 50-56) and a discussion of the role dinoflagellate histones may play in a DVNP-based system has been added to the Discussion (see lines 212-230).

Thirdly, infection of dinoflagellates by the phycodnavirus has never been reported, yet other algal hosts are known, such as *Ectocarpus siliculosus*, and *Feldmannia* sp. Yet no acquisition of this viral gene and subsequent histone displacement have seemed to occur in those hosts. This also raises a question if viral infection can be a sufficient driving force.

These are good points and ones that we have pondered as well. In fact, we did not really mean to imply that viral infection alone was sufficient to drive this process, but rather that it drove the process in the context of the underlying biology of dinoflagellates. Several aspects of dinoflagellate genome biology are unusual, and determining which of these features may have led to others is a puzzle. Our view is that the viral infection was one key, but whether it could drive such change would depend on other aspects of the biology being permissive to that. Infection in other algae has not led to these changes and we do not expect that it should since the biology of the host genome is so different. We have tried to clarify this in the Discussion (see lines 182-185, and 204-211)

To specifically respond to the points made by the reviewer, first, the absence of a dinoflagellate infecting phycodnavirus likely represents the lack of data on identifying these viruses, since relatively few hosts have actually been described. DVNP-homologs have been found in diverse viruses infecting not only stramenopiles but also green algae (see supplemental figure 3c in Shoguchi et al. 2012) . This large taxonomic breadth suggests that there are more DVNP containing phycodnaviruses to be found. This is consistent with the fact that if you analyze environmental metagenomic data, there is a large diversity of DVNP-related proteins in viral metagenomes from around the world, because so much of viral diversity remains uncharacterized. This has been added to the discussion (see lines 186-189).

Secondly, the question of why chromatin divergence has not been reported in other lineages despite exposure to similar viruses is an interesting one, and we have added a section to the Discussion dealing with this (see lines 200-211). As noted above, our position is that the exclusivity of DVNP to the dinoflagellate lineage is because undergoing such dramatic divergence, especially given the toxicity of DVNP, is very unlikely. Dinoflagellates were probably uniquely permissible due to some other unique aspects of their biology that facilitated the histone-to-DVNP transition.

Reviewer #2 (Remarks to the Author):

This is a very clever manuscript by Irwin and colleagues who take the original observation of the mutually independent phylogenetic patterns of DVNPs and histones in dinoflagellates (Current Biology 2012) and ascribe a causality to it using yeast experiments. This is a very key finding. The authors conclude that DVNP expression in yeast is toxic, that it displaces nucleosomes and affects RNA Polymerase recruitment, and most interestingly, that loss of histone genes or histone gene expression can exacerbate these toxic effects.

In many aspects this is the best set of experiments one can perform to describe the mysterious loss of histones and canonical chromatin in dinoflagellates. I find the arguments compelling except for one point that is much more of a discussion point than anything else.

Character displacement (DVNPs for histones) must have proceeded through a deleterious phase if the authors are correct. The more deleterious, the greater the intensity of selection for loss of chromatin genes. But if this were the case, it also holds that the more deleterious, the less likely the event would have occurred at all, or gone to fixation in a species.

We agree with this comment in some senses, but in others think there is an alternative way to imagine it. Specifically, the intermediate state here may not have improved fitness over the initial state, but may be less deleterious compared to its alternative, which is death due to action of the viral protein. In this case, whether the transition was deleterious to the point that the lineage itself competed poorly with competitors would depend on many other unknown factors, such as the infection rate, presence and absence of other coping mechanisms and the extent to which histone depletion would be deleterious. Importantly, the 20% histone reduction in the *spt21Δ* strain of yeast did not have any obvious phenotypic consequences, other than resistance to DVNP (supplemental Fig. 4c). Moreover, histone depletion is harmful largely because of genomic instability and reduced transcriptional control resulting from a loss of suppression of cryptic transcription, and it is possible that different organisms have differing capacities to cope with these issues. Some eukaryotes, such as euglenids, trypanosomes, and intriguingly also dinoflagellates, appear to have a reduced requirement for transcriptional regulation and depend on post-transcriptional control. Lastly, while we agree that deleterious consequences diminish the probability of a transition occurring, we do point out that DVNP-based chromatin has only

been observed in a single eukaryotic lineage. All these points have been added to the discussion (see lines 189-199).

To complete the argument, it might be worth speculating why the DVNP invasion was permitted in spite of its instantly deleterious consequence. After all it is unlikely that the viral invasion would have been complete in the species. As a thought or actual experiment, is it possible for the authors to assess the opposite i.e. is loss of histone gene expression (which is also deleterious in yeast) exacerbated by DVP expression. Could such an event be possibly related to the locking down of the transiently histone-less state in dinoflagellates?

This is related to the point above, and some of our responses are the same. Overall, the question of why DVNP invasion was permitted is interesting and we added a couple possible hypotheses to the Discussion (see lines 172-185).

Firstly, histone depletion could have been an adaptive response to DVNP exposure during infection (the alternative being more deleterious, as noted above) and once resistant, DVNP genes could have been horizontally transferred to dinoflagellates with diminished consequences and been stochastically fixed in the population. Secondly, dinoflagellates could have already had diminished histones at the point at which they were exposed to DVNP. For example, the genome expansion observed in dinoflagellates could have reduced histone abundance through dilution and created a cellular state that was less sensitive to DVNP, thus permitting its retention.

Reviewer #3 (Remarks to the Author):

In this paper, the authors study dinoflagellate-viral-nucleoproteins (DVNPs) interactions in chromatin. The dinoflagellates are unicellular algae and have nuclear chromatin devoid of histone proteins. Histones are replaced by these DVNP proteins which are also relatively small basic protein with >25% lysine residues and about 10% arginine residues, very similar to the basic amino acid content of histones. To study the DVNP protein interactions in chromatin, the authors insert the gene for DVNP into a strain of *Saccharomyces cerevisiae*, express it there, and then assess whether these proteins become chromatin bound and whether they have the properties of chromatin with regular histones.

Figure 1 shows that the DVNP proteins localize to nuclei in yeast cells, but impairs growth. In figure 2, the authors show in a genome wide fashion that DVNPs are associated with the yeast chromatin as complexes similar to nucleosomes by histone displacement. And Figure 3 shows the effect on gene expression of the expression of the DVNPs in yeast cells. Figure 4 shows that there is a histone loss in an adaptive response to DVNP. To interpret this directly, more DVNP, less histone. The big question remains, why? Can the effect be titrated with different proteins? Since DVNP is a basic protein comparable to histones, it is not surprising that these effects are registered in yeast. Maybe an additional control experiment to do, besides TDP-43 which is only

10% basic amino acids, would be to also add in a different equally basic protein such as a protamine, or some other proteins of similar length which are lysine and/or arginine rich.

In terms of why DVNP expression results in histone loss in yeast (Fig. 4c-f), we believe that displaced histones are likely degraded by the Rad53-ubiquitylation pathway which is responsible for degrading excess, unbound, histones in general.

Furthermore, we agree that determining whether the observed effects reflect the specific activity of DVNP or are a ubiquitous result of over expressing a small basic protein is important. Previously we had used TDP-43 to investigate the specificity of the genetic interactions observed with DVNP. However, it is true that some of the biophysical characteristics of DVNP are dissimilar to TDP-43. Therefore, to add to this, we analyzed SGA data examining the genetic interactions of the linker histone, Hho1. Like DVNP, Hho1 is not only toxic when over-expressed but has a very similar isoelectric point and size (Hho1 and DVNP have a P_I of approximately 10 and 11, respectively). Yet despite these similarities, we did not observe the same genetic interactions with HHO1. This data has been included in supplemental Figure 4g and a description of the result has been added (see lines 156-164). Moreover, no effect on histone H3 levels was observed by immunoblot following over-expression of HHO1 (Lawrence et al. 2017 *Genetics*). Therefore, these data further support the specificity of DVNP in triggering histone loss. Protamines are particularly interesting since they replace the majority of histones in sperm and it is possible that the mechanisms of histone displacement by protamines and DVNP could be similar. Therefore, although performing a similar analysis with a protamine would undoubtedly be interesting, the fact that protamines are known to displace histones during spermatogenesis, makes these proteins less than ideal controls for nucleosome displacement by any charged protein.

Minor point:

Page 6: Line 116: Fig 3f should be Fig 4f.

Thank you for noticing this, the text has been changed. Please see line 161.

Reviewer #4 (Remarks to the Author):

The manuscript aims to address the mechanism with which the nucleosomeless status of the dinoflagellates evolved.

1. The description seems to be confused as to whether the data wish to address nucleosomeless or histoneless;

Our study aimed to investigate the evolutionary events that led to the transition between nucleosome and DVNP-based chromatin in dinoflagellates. Therefore, we were particularly interested in the disappearance of bulk nucleosomal chromatin. We have tried to make this

clearer in manuscript (see lines 63-67, and lines 166-167).

2. "Histone loss and histone depletion" was repeatedly mentioned; This is incorrect as there are histone expression in dinoflagellates [ref 28 and Roy and Morse 2012, Plos One <https://doi.org/10.1371/journal.pone.0034340>]

It is correct that complete histone loss has not occurred in dinoflagellates. Many studies have now found evidence for not only the retention but expression of histone genes and we have clarified this by adding a paragraph to the introduction (see lines 50-57). We have also added the Roy and Morse (2012) citation as it provides strong evidence for this. However, dinoflagellates have lost the bulk of their nucleosomal chromatin, and histone proteins are often only barely detected in dinoflagellates despite using sensitive techniques such as mass spectrometry (see Gornik et al. 2012 *Current Biology*, and Roy and Morse 2012 *PLoS ONE*). Therefore, we believe that the notion of "histone depletion" is correct but agree that "histone loss" is ambiguous as some histones remain (probably with massively reduced functional roles in the cell). To correct for this, every instance where "histone loss" was mentioned has now been replaced by more accurate terms, specifically "histone depletion" or "histone reduction".

3. DVNP expression in yeast, and in previous experiment in Toxoplasma, Apparently had no mentioned effect on the genomic copies of histones; therefore the ms did not address the question it set out to address

To reiterate the question, the reviewer is wondering why no mention of the effect DVNP has on genomic copies of histones was made. We were not sure whether the question was specifically addressing histone gene copies or the number of nucleosomes on the genome but below are responses to both interpretations.

If "genomic copies of histones" is referencing histone gene copy number, although the acquisition of DVNP in dinoflagellates does correlate with a loss of histone protein expression, whether or not histone gene copy number is altered in dinoflagellates has not fully been assessed. Marinov and Lynch (2015) examined different histone variants in dinoflagellate transcriptome data but without complete genomic data these analyses can be ambiguous. For example they may reflect different genes or alternatively spliced transcripts and these experiments did not provide insights into the copy number of individual isoforms. Moreover, histone gene copy number varies largely amongst different organisms (for example yeast have two copies of each whereas animals have between 10 and 400 copies) so the effects of histone copy number variation are difficult to interpret. This is partially due to the fact that histone gene expression involves a large amount of post-transcriptional regulation making histone expression not entirely dependent on gene copy number. Lastly, as mentioned above in comment 2, dinoflagellates retain histone genes and this has been clarified in both the Introduction (lines 50-56) and Discussion (lines

212-230). Thus, although histone gene copy number could be an interesting avenue of research once more genomic data is available, it is not pertinent to this study.

However, if "genomic copies of histones" is referring to the number of nucleosomes bound to the yeast genome following DVNP expression, this is shown in Figure 2b-e. These plots reveal a loss of bound nucleosomes following DVNP expression as inferred by MNase sequencing.

Lastly, we must reiterate the question the manuscript set out to address which was how does DVNP interact with canonical nucleosomal chromatin. We provide insights into this question by not only demonstrating the antagonism between DVNP and nucleosomes, but its effects on RNA polymerase and transcription as well as how it genetically interacts with chromatin-related proteins. Therefore, the manuscript did answer the question it set out to address.

A better model could have been in ciliates, as the both apicomplexans and dinoflagellates were evolved from ciliates

In response to this comment we have added a section to the introduction explaining why we chose yeast as a model organism (see lines 61-63).

Although ciliates are more closely related to dinoflagellates than yeast, we believe that yeast is a good model for studying chromatin evolution and was selected over a ciliate such as *Tetrahymena* for a number of reasons. Firstly, yeast have a very well defined chromatin biology and there is a large amount of data available for analysis. These 'baseline' expectations made it easier to detect, analyze, and rationalize perturbations in the system following DVNP expression. None of these expectations are known at this level of detail in ciliates. Secondly, ciliate models lack a number of useful techniques that are available in yeast. For example, the synthetic genetic array (SGA) is not available in ciliates, which would have made the genetic screens much more time consuming and complicated. Thirdly, although ciliates are more closely related to dinoflagellates, their chromatin is not obviously more similar to dinoflagellates than is that of yeast, and other features of their nuclear biology such as their genome organization is in fact more divergent. Thus, we argue that the benefits of using ciliates because of their relatedness to dinoflagellates does not outweigh the benefits gained by using *S. cerevisiae*.

Expression of a DNA-binding protein with higher affinity to DNA binding sites presumably would produce similar effects (e.g. protamines)

As noted in response to reviewer 3, in order to assess whether proteins with a high affinity for DNA binding produce similar effects to DVNP in yeast, we analyzed SGA data from TDP-43 (a foreign DNA-binding protein that is toxic when over-expressed) and have now added an analysis of the linker histone, HHO1 (which has a similar isoelectric point and

size to DVNP, strongly binds DNA, and is also toxic when over-expressed). In both of these cases, we did not see similar effects in terms of histone depletion effecting their toxic phenotypes. This is now shown in supplemental Fig. 4g and described in lines 156-164. The example of protamines is particularly interesting since they replace the majority of histones in sperm and one may hypothesize that the mechanisms of histone displacement by protamines and DVNP could be similar. Unfortunately, data is not currently available describing the effects of expressing protamines in yeast. Regardless, we believe that whether or not proteins similar to DVNP have comparable activities, our conclusions regarding the interactions between DVNP and chromatin remain the same.

REVIEWERS' COMMENTS:

Reviewer #1 (Remarks to the Author):

The revised manuscript has been significantly improved, with most significant comments addressed. I have a few more technical points for clarification.

Line 46: reference 11 cited here was published in 1997, and thus cannot really speak about the function of DVNPs that were only discovered in 2012.

Line 51: the original discovery of genes of all major histones and their modifying proteins in dinoflagellates should be traced back to 2010 ("Spliced leader-based metatranscriptomic analyses lead to recognition of hidden genomic features in dinoflagellates" PNAS 2010 November, 107 (46) 20033-20038).

Lines 69-79: why did you use SV40 nuclear localization signal rather than dinoflagellate native signal? If it was because dinoflagellate signal might not function in the yeast model, then it is inconsistent to infer nucleolar localization based on dinoflagellate nucleolar targeting signal sequence.

Fig. 1C: it seems regardless NLS is in the DVNP protein N- or C terminus, the fusion protein is localized to the nucleus. Does this suggest both dinoflagellate NLS and the yeast NLS in the N-terminus function as nuclear localization signal or that yeast protein NLS can be at both ends for correct nuclear localization? This needs to be clarified.

Line 163: DVNP or TDP-43/Hho1 or both do you mean by "these genetic"?

Reviewer #2 (Remarks to the Author):

I have restricted my comments to the authors' responses to my and the other reviewers' earlier comments. I remain steadfast in my assertion that this is an elegant and indeed unexpectedly clean result about the mutual exclusivity of DVNPs and histone gene expression. Indeed, each of the Reviewers 1-3 identified this as a highly interesting result. Given the 'frozen' evolutionary status of histones in eukaryotes, yeast is a powerful genetic system to test the hypothesis that DVNP expression may have led to the loss of histone gene expression. I see the Results as solid and solidly interpreted.

My comments to the reviewers notwithstanding, its important to point out the distinction between suggestions that might alter the findings or their interpretation or things that alter the scope of the hypothesis that explain these findings. My sense is that the authors had mostly covered their bases for criterion 1 previously and I feel that they have now covered the bases for criterion 2.

Reviewer #3 (Remarks to the Author):

I have read the author's responses to my criticisms of their manuscript and they have adequately addressed all my concerns.

Reviewer #4 (Remarks to the Author):

The authors have done some improvement to the manuscript.
the ex vivo data is interesting

The authors misinterpreted my previous comments as to comparison of DVNPs with protamine;

please comment on why DVNP was not retained in dinoflagellates

biochemical experiments to demonstrate in vitro and in vivo binding properties of DVNP(s) would give strength to the ms,

RESPONSE TO REVIEWERS

Reviewer comments are shown in black, author responses are shown in bolded blue.

Reviewer #1 (Remarks to the Author):

The revised manuscript has been significantly improved, with most significant comments addressed. I have a few more technical points for clarification.

Line 46: reference 11 cited here was published in 1997, and thus cannot really speak about the function of DVNPs that were only discovered in 2012.

It is true that the name DVNP was coined by Gornik et al. (2012). This was done following their investigation into the major basic proteins in the nucleus of the basal dinoflagellate, *Hematodinium*. However an earlier study by Kato et al. (1997), examined the predominant chromatin protein of another basal dinoflagellate, *Oxyrrhis marina*. They dubbed this protein NP23 and provided useful biochemical and cellular localization data. It is now clear that the NP23 protein described by Kato et al. (1997) corresponds to DVNP and this was also acknowledged by Gornik et al. (2012). Therefore we believe the inclusion of the 1997 citation is justified.

Line 51: the original discovery of genes of all major histones and their modifying proteins in dinoflagellates should be traced back to 2010 (“Spliced leader–based metatranscriptomic analyses lead to recognition of hidden genomic features in dinoflagellates” PNAS 2010 November, 107 (46) 20033-20038).

This citation has now been added. Please see reference 13.

Lines 69-79: why did you use SV40 nuclear localization signal rather than dinoflagellate native signal? If it was because dinoflagellate signal might not function in the yeast model, then it is inconsistent to infer nucleolar localization based on dinoflagellate nucleolar targeting signal sequence.

The SV40 nuclear localization signal was added to ensure that DVNP would be nuclear localized even if the dinoflagellate signal was not sufficient. When we analyzed the localization of DVNP we found that it was within the nucleus but also appeared to show some degree of nucleolar localization. We refrained from speculating to much about why this was as it was not pertinent to our study. However, we hypothesized that the basic N-terminus of DVNP may be acting as a general nucleolar targeting signal, as opposed to there being a dinoflagellate-specific nucleolar targeting signal. This has been clarified in lines 84-86.

Fig. 1C: it seems regardless NLS is in the DVNP protein N- or C terminus, the fusion protein is localized to the nucleus. Does this suggest both dinoflagellate NLS and the yeast NLS in the N-terminus function as nuclear localization signal or that yeast protein NLS can be at both ends for correct nuclear localization? This needs to be clarified.

We found that DVNP was nuclear localized regardless of whether the NLS was absent or present at the N- or C-terminus. Nuclear localization signals can function at both ends of the protein but we agree with the reviewer that this could also indicate that the native dinoflagellate NLS facilitates nuclear trafficking. However, DVNP is also a small protein and thus may be able to passively diffuse through the nuclear pore complex into the nucleus. Therefore, this observation can be explained by a number of possibilities and because it is not pertinent to our study we have refrained from speculating. Regardless we have now added clarification to this point (see lines 82-84).

Line 163: DVNP or TDP-43/Hho1 or both do you mean by “these genetic”?

By "these" we were referring to the genetic interactions of DVNP. This has now been corrected (please see line 170).

Reviewer #2 (Remarks to the Author):

I have restricted my comments to the authors' responses to my and the other reviewers' earlier comments. I remain steadfast in my assertion that this is an elegant and indeed unexpectedly clean result about the mutual exclusivity of DVNPs and histone gene expression. Indeed, each of the Reviewers 1-3 identified this as a highly interesting result. Given the 'frozen' evolutionary status of histones in eukaryotes, yeast is a powerful genetic system to test the hypothesis that DVNP expression may have led to the loss of histone gene expression. I see the Results as solid and solidly interpreted.

My comments to the reviewers notwithstanding, its important to point out the distinction between suggestions that might alter the findings or their interpretation or things that alter the scope of the hypothesis that explain these findings. My sense is that the authors had mostly covered their bases for criterion 1 previously and I feel that they have now covered the bases for criterion 2.

Thank you for your comments.

Reviewer #3 (Remarks to the Author):

I have read the author's responses to my criticisms of their manuscript and they have adequately addressed all my concerns.

Thank you for your comments.

Reviewer #4 (Remarks to the Author):

The authors have done some improvement to the manuscript.
the ex vivo data is interesting

The authors misinterpreted my previous comments as to comparison of DVNPs with protamine;

We apologize for our misinterpretation.

please comment on why DVNP was not retained in dinoflagellates

DVNP was actually retained in dinoflagellates and appears to have replaced histones as the predominant DNA packaging protein. However if the reviewer meant to ask why DVNP was retained in dinoflagellates the answer becomes more complicated. It's possible that in a histone-depleted system, DVNP was capable of promoting genome stability and repressing cryptic transcription. We have mentioned this briefly in lines 187-189. However, rigorously testing why DVNP was retained will require a comprehensive characterization of the chromatin of dinoflagellates and their close relatives and *in vivo* experiments in dinoflagellates themselves. Therefore this is beyond the scope of the current manuscript which aimed to understand the reasons for histone depletion and we wish to refrain from speculating too much on this subject. Understanding the reasons for why DVNP was retained will be an interesting avenue for future research.

biochemical experiments to demonstrate *in vitro* and *in vivo* binding properties of DVNP(s) would give strength to the ms,

Previous studies by Gornik et al. (2012) have already performed *in vitro* binding experiments with DVNP in order to assess its affinity for DNA. Their results showed that DVNP is capable of strong non-specific DNA binding. In agreement with this result, our *in vivo* binding experiments (ex. DVNP ChIP-seq) showed that DVNP binds the yeast genome. We have acknowledged this in lines 96-101. Future experiments exploring the biophysical properties of DNA binding by DVNP will be interesting.